# Optimization of Reinforced Concrete Sections under Compression and Biaxial Bending by Using a Parallel Firefly Algorithm

**Gregorio Sánchez-Olivares * and Antonio Tomás** 

Department of Mining and Civil Engineering, Universidad Politécnica de Cartagena (UPCT), Paseo Alfonso XIII 52, 30203 Cartagena, Murcia, Spain; antonio.tomas@upct.es
* Correspondence: gregorio.sanchez@upct.es; Tel.: +34-968-325-927

**Abstract:** A simple formulation for the optimal design of reinforced concrete sections under compression and biaxial bending was established in a previous work by the authors. In that work, it was found that the formulation produced satisfactory results when used together with three algorithms that belong to the nature-inspired meta-heuristic algorithm group. However, despite the favorable results obtained, the necessary calculation times were extensive in all the cases. In order to solve this problem, the authors implemented a parallel calculation strategy in the algorithm that gave better results in the previous work. It was possible to verify, through two examples, that this strategy reduces calculation times as more processes are used in parallel, and that the adjustments made in the algorithm favor reaching designs close to the global optimum independently of the number of parallel processes adopted.

**Keywords:** optimization; parallel algorithm; meta-heuristic methods; reinforced concrete sections; biaxial loading

## 1. Introduction

In recent years, meta-heuristic algorithms have been developed to solve not only simple engineering problems but also complex ones that present nonlinear objective functions and numerous nonlinear constraints. Biologically-inspired algorithms belong to the nature-inspired meta-heuristic algorithm group. Those algorithms can be divided into two categories: evolution-based and swarm-based methods [1]. The best-known and most used evolution-based algorithms are Genetic Algorithms (GA) [2,3]. The most popular of the Swarm-based methods is Particle Swarm Optimization (PSO), which is inspired by the group behavior of flocks of birds in flight [4].

Physics-based meta-heuristic algorithms, another type of nature-inspired algorithm, are based on the physical laws of the universe. The most popular of these is Simulated Annealing (SA) [5]. Besides these algorithms, there are human-based algorithms, the most popular of which is Harmony Search (HS) [6].

At the same time as these new meta-heuristic algorithms were developed, a number of "no free lunch" (NFL) theorems were presented [7]. These NFL theorems establish that for any algorithm, any elevated performance in one type of problem is offset by performance in another type. Therefore, after a period of proposing new algorithms, some of them began to be applied and fitted to specific engineering problems. The optimum design of concrete structures, which has been widely used in building [8], road [9], rail [10], and marine [11] engineering, is one of these specific problems.

In parallel to this period dedicated to the development and application of new algorithms, studies have been oriented toward detecting differences in their performance and some improvements have been proposed. PSO has been shown to be more efficient than GAs [12,13]. The work by Valvano et al. [14] is oriented toward the development of a new,

accurate, and efficient Decline Population Swarm Optimization ($P_D$SO) algorithm. This new method, applied to vibro-acoustic problems, has resulted in increased performance of the standard PSO. Moreover, the Firefly Algorithm (FA), which is a swarm-based method, has been shown to be more efficient than GAs and PSO [15]. The work by Gandomi et al. [16] shows how the use of different chaotic systems to replace the parameters of the FA improves the efficiency of the algorithm. It is also necessary to mention the work by Fister et al. [17], which shows the widespread use of the FA in various fields of engineering. The works by Gandomi et al. [18,19] and Talatahari et al. [20] are particularly interesting.

In the case of complex structural systems, the number of variables and constraints increases considerably, so the chosen methodology must be based on efficient algorithms in order to reduce the computational cost [21,22]. However, less effort has been made to solve structural system problems where elements subject to biaxial bending appear. It is necessary to keep in mind that under biaxial bending, the computational cost increases if the equivalent rectangular compressive stress block is not used for calculating stresses in the compressed zone of the section [23]. Thus, the need to reduce costs is increasing constantly, and the use of efficient algorithms, such as the FA, in these types of problems, is essential [24]. Many recent methods are efficient, although not all of them respond well to real problems with constraints [22].

Meta-heuristic algorithms need a long time to achieve a final design. Parallel computing strategies sharing the computation load over several processors have started to be applied to reduce this computational time. Thierauf and Cai [25] present a parallel-evolution method for structural optimization. Firstly, the problem is divided into two subproblems; one with discrete design variables and the other with continuous ones. Then, each subproblem is solved using a parallel evolution approach. Leite and Topping [26] review and evaluate different parallel schemes applied to an SA algorithm, where the design dominium is complex and very constrained, and the evaluation of the objective functions can result in medium to high computational costs. In the conclusion, the authors warn that parallelization seems to be the only general strategy able to reduce time and open its applicability to engineering design. Hasançebi et al. [27] address an evolutionary meta-heuristic algorithm using an integrated parallelization technique for the structural optimization of high-rise steel buildings. The parallelization is based on a master–slave model that provides optimal solutions with a reduction of time and without a lack of accuracy. Truong et al. [28] present an approach integrating GAs and OpenMP applied to steel structures with semi-rigid connections that produce a clear reduction in computational time.

The previous formulation proposed by the authors [29] for the optimal design of reinforced concrete sections under compression and biaxial bending includes design considerations in accordance with the Eurocode 2 (EC2) [30] or ACI318 [31] standards and can be implemented in any meta-heuristic algorithm. In this paper, the authors implemented and checked a parallel FA, where the formulation is considered, with the aim of reducing computational time in the process. The two problems solved in the previous work by the authors [29] now show not only that the parallel strategy reduces the time needed to arrive at a solution but also produces good results regardless of the number of processors used.

## 2. Optimization of Concrete Rectangular Cross-Sections

### 2.1. Optimum Design Problem

The problem of calculating the optimum geometry and reinforcement in a rectangular concrete cross-section subjected to biaxial bending and axial force may be generally stated as:

To find the design variable vector

$$\boldsymbol{X}(X_1, X_2, \ldots, X_{nv}) \tag{1}$$

to minimize the objective function

$$f(\boldsymbol{X}) \tag{2}$$

subject to the constraints

$$g_c(X) \geq 0; c = 1, 2, \ldots, nc \tag{3}$$

$$X_v^L \leq X_v \leq X_v^U; v = 1, 2, \ldots, nv \tag{4}$$

where $X$ is the $nv$-dimensional design variable vector; $f(X)$ is the objective function; $g_c(X)$ is the inequality constraint $c$; $X_v^L$ is the lower bound for the variable $v$ and $X_v^U$ is the upper bound for the variable $v$; $nc$ is the total number of constraints, and $nv$ is the number of variables.

The FA reproduces the social behavior of fireflies in a simple and idealized way. Fireflies communicate with each other, seek prey, and find a partner using different patterns of bioluminescent flashes. The characteristics of these flashes are idealized to achieve the development of this algorithm. In a similar way to how real fireflies fly through three-dimensional space, modifying their spatial coordinates, the fireflies of the algorithm, or individuals, now fly through $nv$-dimensional space where the coordinates are now the variables of the design problem (1). The best design corresponds to an individual that flashes more than the others because it is in a position that minimizes a function, called the fitness function. This fitness function takes into account the objective function (2) and the inequality constraints (3) that are violated ($g_c(X) < 0$) together. During the process, the algorithm ensures that the variables are within the bounds (4) that define the design space.

*2.2. Variables*

There are twelve design variables that were considered for each individual $i$ (Figure 1): the depth of neutral axis $z_i$; the angle of the neutral fiber $\theta_i$; the width $b_i$; the height $h_i$; the bar diameter of the right side reinforcement $\varphi'_{y,i}$; the number of bars on the right side of the section $n'_{y,i}$; the bar diameter of the left side reinforcement $\varphi_{y,i}$; the number of bars on the left side of the section $n_{y,i}$; the bar diameter of the top side reinforcement $\varphi'_{x,i}$; the number of bars on the top side of the section $n'_{x,i}$; the bar diameter of the bottom side reinforcement $\varphi_{x,i}$; and the number of bars on the bottom side of the section $n_{x,i}$.

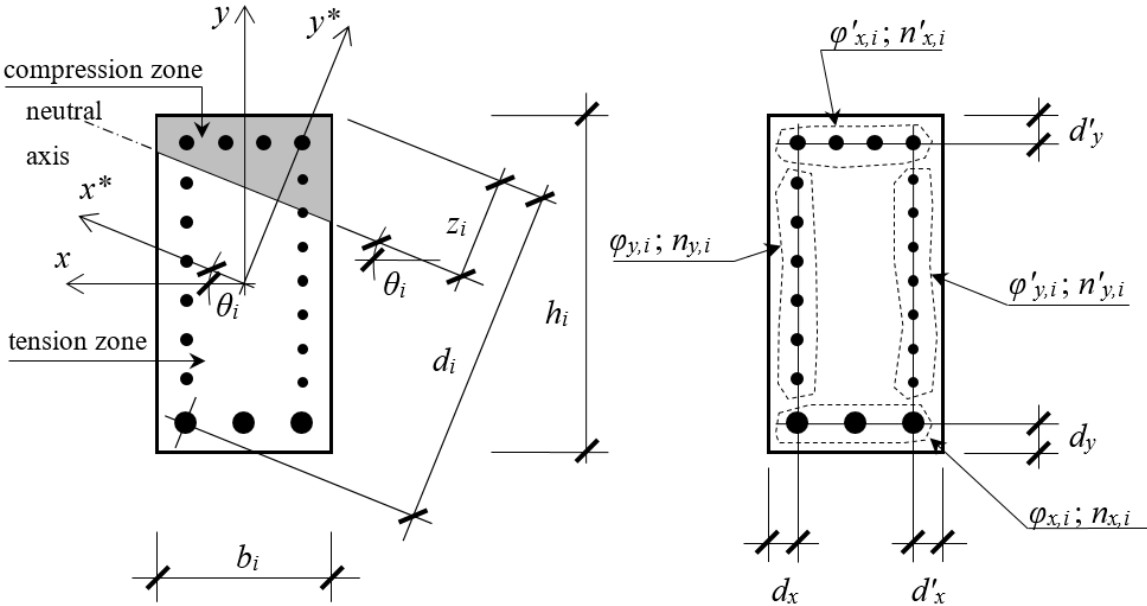

**Figure 1.** Cross-section geometry and design variables for the individual $i$.

The design variables $z_i$ and $\theta_i$ are continuous, and the rest of them are discrete. The designer can define limits for these variables. The variables $b_i$ and $h_i$ adopt values of 5 by 5 mm. The variables $n'_{y,i}$, $n_{y,i}$, $n'_{x,i}$ and $n_{x,i}$ take integer values. Finally, the variables $\varphi'_{y,i}$, $\varphi_{y,i}$, $\varphi'_{x,i}$ and $\varphi_{x,i}$ adopt values corresponding to commercial diameters of reinforcing steel bars.

The proposed formulation considers optimization variables instead of design variables. The design variables $\theta_i$, $b_i$, $h_i$, $\varphi'_{y,i}$, $n'_{y,i}$, $\varphi_{y,i}$, $n_{y,i}$, $\varphi'_{x,i}$, $n'_{x,i}$, $\varphi_{x,i}$, $n_{x,i}$ were normalized by dividing them by their upper bounds in order to obtain the optimization variables: $\zeta_i$, $\beta_i$, $\eta_i$, $\omega'_{y,i}$, $\nu'_{y,i}$, $\omega_{y,i}$, $\nu_{y,i}$, $\omega'_{x,i}$, $\nu'_{x,i}$, $\omega_{x,i}$, $\nu_{x,i}$, respectively. Instead of the design variable $z_i$, the optimization variable $\xi_i$ is considered as follows:

$$\xi_i = \frac{2}{\pi}\arctan(z_i) \tag{5}$$

Thereby, the optimization variables adopt values within the same order of magnitude. This ensures that the variables will have the same relative importance when used in the design algorithm, thus favouring efficiency.

*2.3. Fitness Function*

The fitness function proposed is as follows:

$$F(\boldsymbol{X}_i) = \frac{f(\boldsymbol{X}_i)}{f(\boldsymbol{X}^U)} - \sum_{j=1}^{mvi} g_j(\boldsymbol{X}_i)\left(\frac{f(\boldsymbol{X}_i)}{f(\boldsymbol{X}^U)} + \frac{f(\boldsymbol{X}^U)}{f(\boldsymbol{X}_i)}\right) \tag{6}$$

where $\boldsymbol{X}_i = (\xi_i, \zeta_i, \beta_i, \eta_i, \omega'_{y,i}, \nu'_{y,i}, \omega_{y,i}, \nu_{y,i}, \omega'_{x,i}, \nu'_{x,i}, \omega_{x,i}, \nu_{x,i})$ is the 12-dimensional optimization variable vector for the individual $i$; $\boldsymbol{X}^U$ is the 12-dimensional optimization variable upper bound vector; $f(\boldsymbol{X}_i)$ is the objective function to be minimized; and *mvi* is the number of violated constraints ($g_j < 0$) for the individual $i$.

*2.4. Objective Function*

The objective function $f_i$ of the optimization problem for the individual $i$ is the cost of RC cross-section per unit length

$$f(\boldsymbol{X}_i) = f_i = P_c A_{c,i} + P_s \rho_s A_{st,i} + P_f L_{f,i} \tag{7}$$

where $P_c$ is the price of concrete per unit volume; $P_s$ is the price of steel per kilogram; $\rho_s$ is the density of steel and $P_f$ is the price of formwork per unit area; $A_{c,i}$ is the area of the concrete cross-section for the individual $i$; $A_{st,i}$ is the total area of steel reinforcement for the individual $i$; and $L_{f,i}$ is the perimeter of form for the individual $i$.

*2.5. Constraints*

2.5.1. Reinforcement Constraints

The normalized reinforcement constraints are the following:

$$g_{As,i} = \frac{A_{s,i}}{A_{s,min,i}} - 1 \geq 0 \tag{8}$$

$$g_{A's,i} = \frac{A'_{s,i}}{A'_{s,min,i}} - 1 \geq 0 \tag{9}$$

$$g_{Ast,i} = \frac{A_{st,i}}{A_{st,min,i}} - 1 \geq 0 \tag{10}$$

where $A_{s,i}$ is the amount of steel in the tension reinforcement for the individual $i$; $A_{st,min,i}$ is the minimum amount of steel in the tension reinforcement allowed for the individual $i$; $A'_{s,i}$ is the amount of steel in the compression reinforcement for the individual $i$; $A'_{s,min,i}$ is the minimum amount of steel in the compression reinforcement allowed for the individual $i$; $A_{st,i}$ is the total area of reinforcement for the individual $i$; and $A_{st,min,i}$ is the minimum total area of reinforcement allowed for the individual $i$.

### 2.5.2. Ductility Constraint

The normalized form of the ductility constraint for the individual *i* is

$$g_{duct,i} = 1 - \frac{z_i}{z_{max,i}} \geq 0 \tag{11}$$

where $z_{max,I}$ is the maximum neutral axis depth for the individual *i*.

### 2.5.3. Steel Reinforcement Spacing Constraints

The normalized form of the spacing constraints for the individual *i* are

$$g_{sp,min,i} = \frac{s_i}{s_{min,i}} - 1 \geq 0 \tag{12}$$

$$g_{sp,max,i} = 1 - \frac{s_i}{s_{max,i}} \geq 0 \tag{13}$$

where $s_i$ is the spacing between the steel bars of reinforcement for the individual *i*; $s_{min,i}$ is the minimum spacing for the individual *i*; and $s_{max,i}$ is the maximum spacing closest to the tension faces for the individual *i*.

### 2.5.4. Strength Constraints

Two strength constraints are defined for the individual *i* to ensure that the cross-section withstands the design loads. The first constraint assures sufficient strength against the combined flexure and axial load, the second one ensures that the load eccentricity is the same as the strength eccentricity of the cross-section. The first constraint in normalized form for the individual *i* is:

$$g_{s1,i} = 1 - \frac{\sqrt{P_u^2 + M_{x,u}^2 + M_{y,u}^2}}{\varphi(\mathbf{X}_i)\sqrt{P_n(\mathbf{X}_i)^2 + M_{x,n}(\mathbf{X}_i)^2 + M_{y,n}(\mathbf{X}_i)^2}} =$$
$$1 - \frac{\sqrt{P_u^2 + M_{x,u}^2 + M_{y,u}^2}}{\varphi_i\sqrt{P_{n,i}^2 + M_{x,n,i}^2 + M_{y,n,i}^2}} \geq 0 \ \ (\text{ACI 318}) \tag{14}$$

or

$$g_{s1,i} = 1 - \frac{\sqrt{N_{Ed}^2 + M_{x,Ed}^2 + M_{y,Ed}^2}}{\sqrt{N_{Rd}(\mathbf{X}_i)^2 + M_{x,Rd}(\mathbf{X}_i)^2 + M_{y,Rd}(\mathbf{X}_i)^2}} =$$
$$1 - \frac{\sqrt{N_{Ed}^2 + M_{x,Ed}^2 + M_{y,Ed}^2}}{\sqrt{N_{Rd,i}^2 + M_{x,Rd,i}^2 + M_{y,Rd,i}^2}} \geq 0 \ \ (\text{EC 2}) \tag{15}$$

where $P_u$ is the factored axial force; $M_{x,u}$ is the factored moment about *x*-axis at cross-section; $M_{y,u}$ is the factored moment about *y*-axis at cross-section; $\varphi_I$ is the strength reduction factor for the individual *i*; $P_{n,i}$ is the nominal axial load normal to the cross-section for the individual *i*; $M_{x,n,i}$ is the nominal moment about *x*-axis at the cross-section for the individual *i*; $M_{y,n,i}$ is the nominal moment about *y*-axis at the cross-section for the individual *i*; $M_{x,Ed}$ is the design value of the applied internal flexural moment about *x*-axis at the cross-section; $M_{y,Ed}$ is the design value of the applied internal flexural moment about *y*-axis at the cross-section; $N_{Rd,I}$ is the design axial resistance of the cross-section for the individual *i*; $M_{x,Rd,i}$ is the design moment resistance of the cross-section about *x*-axis for the individual *i*; $M_{y,Rd,i}$ is the design moment resistance of the cross-section about *y*-axis for the individual *i*.

The second constraint in normalized form for the individual *i* is

$$g_{s2,i} = -\left[\frac{(1 - cos\psi_i)}{2}\right]^{\frac{1}{q}}; q = \left(e^{\frac{cos\psi_i}{C_1}} - \frac{cos\psi_i}{C_2}\right); \begin{cases} C_1 \in [1,\ 2] \\ C_2 \in [5,\ 50] \end{cases} \tag{16}$$

with

$$cos\psi_i = \frac{P_u}{\sqrt{P_u^2+M_{x,u}^2+M_{y,u}^2}}\frac{P_{n,i}}{\sqrt{P_{n,i}^2+M_{x,n,i}^2+M_{y,n,i}^2}}$$
$$+\frac{M_{x,u}}{\sqrt{P_u^2+M_{x,u}^2+M_{y,u}^2}}\frac{M_{x,n,i}}{\sqrt{P_{n,i}^2+M_{x,n,i}^2+M_{y,n,i}^2}} \quad (17)$$
$$+\frac{M_{y,u}}{\sqrt{P_u^2+M_{x,u}^2+M_{y,u}^2}}\frac{M_{y,n,i}}{\sqrt{P_{n,i}^2+M_{x,n,i}^2+M_{y,n,i}^2}} \quad (\text{ACI 318})$$

or with

$$cos\psi_i = \frac{N_{Ed}}{\sqrt{N_{Ed}^2+M_{x,Ed}^2+M_{y,Ed}^2}}\frac{N_{Rd,i}}{\sqrt{N_{Rd,i}^2+M_{x,Rd,i}^2+M_{y,Rd,i}^2}}$$
$$+\frac{M_{x,Ed}}{\sqrt{N_{Ed}^2+M_{x,Ed}^2+M_{y,Ed}^2}}\frac{M_{x,Rd,i}}{\sqrt{N_{Rd,i}^2+M_{x,Rd,i}^2+M_{y,Rd,i}^2}} \quad (18)$$
$$+\frac{M_{y,Ed}}{\sqrt{N_{Ed}^2+M_{x,Ed}^2+M_{y,Ed}^2}}\frac{M_{y,Rd,i}}{\sqrt{N_{Rd,i}^2+M_{x,Rd,i}^2+M_{y,Rd,i}^2}} \quad (\text{EC 2})$$

*2.6. Optimization Methodology*

2.6.1. Firefly Algorithm (FA)

Highly non-linear optimization problems are common in structural design. Most of them include many variables that can be discrete and/or continuous, and complex, non-linear constraints [24]. Among the advantages of the FA is that it efficiently overcomes these problems [1,32]. In the previous formulation, a sample of non-linear functions (objective function, fitness function, and constraints) was proposed.

The FA is a swarm-based method whose search process depends on two combined strategies: exploration and exploitation. The former focuses on searching for the global optimum, whereas the latter focuses on selecting the best design found thus far. During the process of design, the parameters of the algorithm allocate more weight to one of the strategies than to the other.

The FA ideally reproduces the behavior of fireflies. Three rules are considered:

(i)   All the fireflies in a population have just one gender and any of them can be attracted to another.
(ii)  The attraction between two fireflies in the whole population is directly proportional to the brightness of their luminescence. This attraction lessens when distances increase. Observing a pair of fireflies, the less bright one moves toward the brighter one.
(iii) The brightness of a specific firefly is linked to the value of its fitness function.

The attraction $\beta$ between two fireflies is

$$\beta(r) = \beta_0 \frac{1}{1+\gamma r^m} \quad (19)$$

where $\gamma$ is the absorption coefficient for a given medium; $m$ is an integer; $r$ is the distance between these two fireflies; and $\beta_0$ is the attraction at $r = 0$.

The movement of a firefly $i$, which is attracted to another, brighter firefly $j$, is expressed by the following expression:

$$p_i^{k+1} = p_i^k + \left(p_j^k - p_i^k\right)\beta_0 \frac{1}{1+\gamma r_{ij}^m} + \alpha\left(rand - \frac{1}{2}\right)\left(X^U - X^L\right) \quad (20)$$

where $k$ is the current iteration; $p_i^k$ is the spatial coordinate vector of the $i$-th firefly at the $k$-th iteration; $p_j^k$ is the spatial coordinate vector of the $j$-th firefly at the $k$-th iteration; $\alpha$ is the randomization parameter; rand is a random number generator uniformly distributed in [0, 1]; $X^L$ is the optimization variable lower bound vector; and $r_{ij}$ is the Cartesian distance

$$r_{ij} = \|p_i^k - p_j^k\| \quad (21)$$

The potential oscillatory behavior of the algorithm can be avoided by reducing the randomization parameter $\alpha$ as the process progresses. Research by Gandomi et al. [18] has established that $\alpha \in [0.01 \, , \, 1]$ can be taken.

The absorption coefficient $\gamma$ controls the speed of the process convergence. If $\gamma$ is near zero, the attraction is constant, and each firefly can be seen by the whole population. However, if $\gamma$ is large, the attraction decreases and the fireflies become almost blind. This situation is as if they were flying in the fog, with almost random movement.

2.6.2. Modified Version (MPFA) of the Firefly Algorithm

(i)     Parallelization and migration.

The authors included a parallel computing strategy based on dividing the whole population into subpopulations. Each subpopulation searches for the optimum design of the best solution found in the subpopulation up to that point, taking exploration and exploitation into account. A code was implemented in MATLAB® (see the Appendix A) to address the parallel strategy using the `parfor` MATLAB® command to distribute the work among processors [33].

Along with the parallel computing strategy, another strategy is needed to enable the firefly subpopulations to communicate with each other, so that the information that resides in each subpopulation can be shared by the rest of the subpopulations. In this way, the authors prevent the parallel computing strategy from merely dividing the population into subpopulations that seek the optimal design independently.

A way to do this is by migrating fireflies among subpopulations. The migration process must ensure that information passes among subpopulations without bias, in order not to force the search in a specific direction. The migration strategy implemented is based on several steps: (i) the one with the best of all the designs achieved up to that point is chosen as the origin subpopulation; (ii) individuals from the origin subpopulation are chosen randomly; (iii) destination subpopulations are chosen randomly, and (iv) the individuals of the destination subpopulations are substituted for the individuals of the origin subpopulation.

As the origin subpopulation is the one that has the best design of those found up to that point in the process of searching for the global optimum, there is a bias in the information that is transmitted. In fact, if the origin subpopulation is close to a local optimum, then the movement of the fireflies in that direction is forced by bringing individuals who are close to that local optimum to other subpopulations. However, this bias does not occur due to the second modification introduced in the FA, which is as follows.

(ii)     Small random displacements.

Two small random displacements on the variables $z$ and $\theta$ are developed; one for the brightest firefly of each subpopulation (local, small, random displacement, LSRD), and another for the brightest firefly of the whole population (global, small, random displacement, GSRD).

$$z^{k+1} = z^k + \alpha^2 \left( rand - \frac{1}{2} \right) \left( z^U - z^L \right) \tag{22}$$

$$\theta^{k+1} = \theta^k + \alpha^2 \left( rand - \frac{1}{2} \right) \left( \theta^U - \theta^L \right) \tag{23}$$

These LSRD and GSRD displacements are essential to improve the performance of the algorithm as was verified by the authors after running many examples, some of which are included in this work.

This improvement is due to the fact that LSRD does not only increase the firefly's mobility but also that of the whole subpopulation that follows its movements. Additionally, the brightest firefly of the subpopulation may cease to be so due to this LSRD. Furthermore, it is possible that the subpopulation considered as the origin is no longer so because the brightest firefly may now belong to another subpopulation. Finally, GSRD is necessary when a high number of subpopulations are considered because of the low number of

fireflies in each subpopulation, which reduces the effect of LSRD. As a consequence, the combined effect of both LSRD and GSRD helps biases disappear and exploration to improve, decreasing the probability of the process ending at a local minimum.

These basic operations were implemented in MATLAB®and are summarized in Figure 2.

```
begin
  Define size of population NF and number of subpopulations np
  parfor l = 1 : np
    Generate random subpopulation l [X¹₁, X¹₂, ..., X¹ᵢ, ..., X¹_NF/np]
  end for l
  for k = 1 : kₘₐₓ
    Evaluate randomization parameter α(k/kₘₐₓ)
    Evaluate absorption coefficient γ(k/kₘₐₓ)
    parfor l = 1 : np
      Evaluate fitness functions F¹ᵢ
      Find the current best min([F¹₁, F¹₂, ..., F¹ᵢ, ..., F¹_NF/np])
      Rank the fireflies [F¹₁, ..., F¹_NF/np]=[F¹_MIN, ..., F¹_MAX]
      for i = NF/np : -1 : 2
        for j = i-1 : -1 : 1
          if F¹ᵢ > F¹ⱼ
            Evaluate Cartesian distance r¹ᵢⱼ
            Evaluate attractiveness β¹ᵢ
            Move individual i towards individual j
            Evaluate F¹ᵢ
            if F¹ᵢ < F¹₁
              Individual X¹ᵢ is the current best (X¹₁)
              Rank the fireflies [F¹₁, ..., F¹_NF/np]=[F¹_MIN, ..., F¹_MAX]
              Move individual X¹₁ randomly (LSRD)
            end if
          end if
        end for j
      end for i
    end for l
    Find the best design X₁ among all subpopulations
    Move individual X₁ randomly (GSRD)
    Migrate individuals randomly
  end for k
  Post-process results and visualization
end
```

**Figure 2.** Pseudo-code of modified version (MPFA).

### 3. Examples

#### 3.1. Cross-Section under Flexure

A cross-section under a factored flexural moment $M_{x,u}$ = 400 kNm is studied. The strength of materials are $f_c'$ = 30 MPa for concrete and $f_y$ = 500 MPa for steel. The clear cover of reinforcement is 40 mm and the stirrup diameter is 10 mm. The cost is calculated using a price of $P_c$ = 100 €/m³ for concrete, $P_s$ = 1.2 €/kg for steel and $P_f$ = 30 €/m² for formwork. The length of form is $b + 2h$. The optimization variable $\xi$ is constrained in the interval [−0.7, 0.7]; the angle of the neutral fiber $\theta$ in [0, $\pi/2$] rad; the width of the cross-section $b$ in [0.30, 0.50] m; the height $h$ in [0.30, 0.90] m; the reinforcement bar diameters $\varphi'_x$, $\varphi_x$ in [10, 32] mm; the number of reinforcement bars $n'_y$, $n_y$ in [0, 0] and $n'_x$, $n_x$ in [2, 10]. The constraints (8) to (10), (11), (12) to (15) and (16) are considered in this example. The parameters of the Equation (16) are $C_1$ = 2 and $C_2$ = 5. The MPFA parameters were already

tuned for this example, as can be seen in [29]. The values adopted in this work are $\beta_0 = 1$, $m = 2.0$, $\gamma_0 = 10$, $\alpha_0 = 1$, $NF = 300$, $k_{max} = 1000$ and $t_c = 0.1$.

The best solution obtained from 50 runs using MPFA with eight subpopulations is considered to be the optimal design solution. The optimal design solutions obtained considering ACI318 and EC2 standards are shown in Table 1.

**Table 1.** Optimal design solutions obtained using the MPFA.

|                    | **ACI318** | **EC2** |
|--------------------|------------|---------|
| $z$(mm)            | 152.3      | 136.3   |
| $\theta$(rad)      | 0          | 0       |
| $b$(mm)            | 300        | 325     |
| $h$(mm)            | 585        | 605     |
| $\varphi'_x$(mm)   | 10         | 10      |
| $n'_x$             | 2          | 4       |
| $\varphi_x$(mm)    | 32         | 20      |
| $n_x$              | 3          | 6       |
| Cost (€/m)         | 85.54      | 86.19   |

The average time of 50 runs, considering ACI318 standard and using MPFA with different numbers of subpopulations, was obtained. Although there is a difference in the average time depending on the computer used, the average time was considered to work out the metric "speed-up", which refers to how much a parallel FA is faster than a sequential FA [34]. The speed-up is shown in Figure 3. "Efficiency", which is the ratio of speed-up and the number of processors used to solve the example, measures how efficiently the MPFA uses the parallel resources [34]. Efficiency is shown in Figure 4.

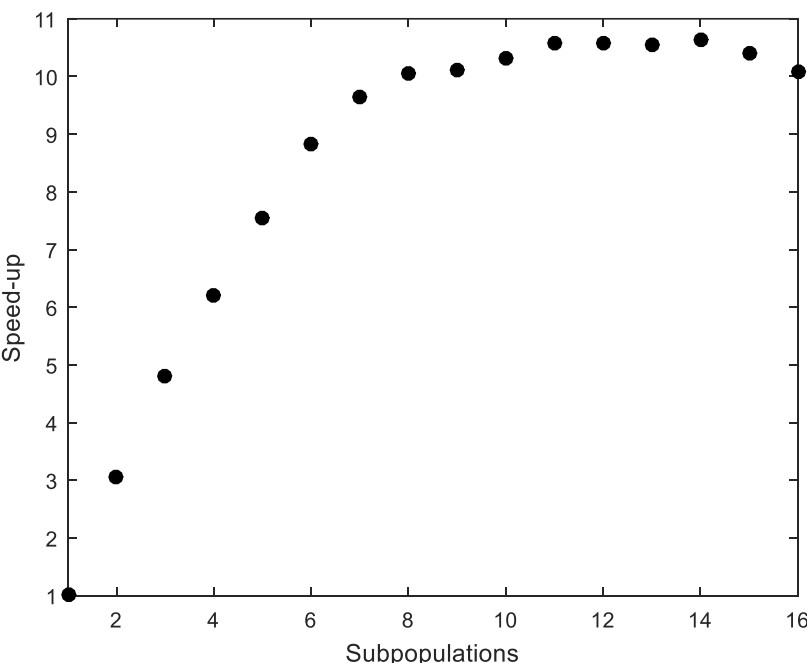

**Figure 3.** Cross-section under flexure. Speed-up.

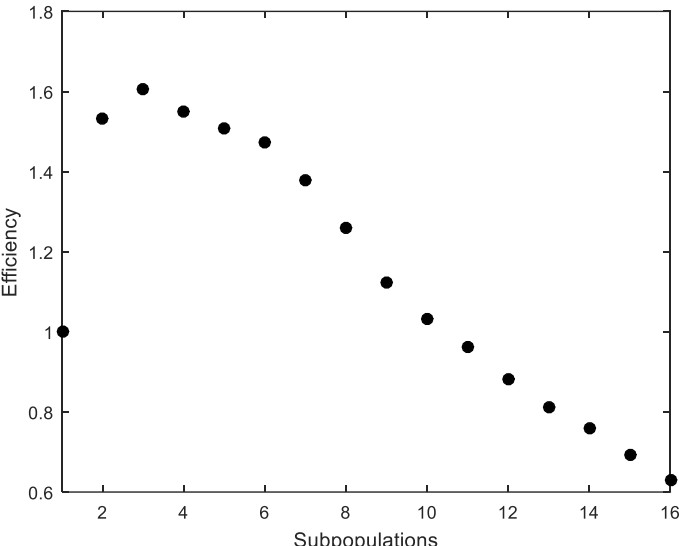

**Figure 4.** Cross-section under flexure. Efficiency.

The parallelization produces satisfactory results despite its simple implementation (Figure 3). However, the efficiency quickly drops with more than four processors (Figure 4). The computer that was used to run the calculations is a workstation equipped with an Intel Xeon CPU E5-1620 v2 3.70 GHz with eight cores.

"Accuracy", which is the mean value of several independent runs when compared to a reference value, measures how efficiently the MPFA uses the parallel resources [34]. "Robustness" is linked to the scatter of obtained results [34] (for example, standard deviation) for several independent runs of MPFA.

The mean cost, accuracy (reference value of Cost = 85.54, from Table 1), and robustness (standard deviation) from 50 runs considering ACI318 standard and using MPFA with a different number of subpopulations were obtained. These metrics are shown in Tables 2–4, considering only LSRD, only GSRD, and both LSRD and GSRD, respectively. For comparison purposes, the average values of the performance metrics were also obtained, and they are shown in the last row of Tables 2–4.

**Table 2.** Cross-section under flexure. Local, small, random displacement (LSRD). Performance metrics.

| Subpopulations | Mean Cost | Accuracy | Robustness |
|:---:|:---:|:---:|:---:|
| 1 | 85.81 | 0.9969 | 0.51 |
| 2 | 85.83 | 0.9966 | 0.52 |
| 3 | 85.78 | 0.9972 | 0.76 |
| 4 | 85.68 | 0.9984 | 0.35 |
| 5 | 85.86 | 0.9963 | 0.81 |
| 6 | 85.68 | 0.9983 | 0.36 |
| 7 | 85.76 | 0.9974 | 0.47 |
| 8 | 85.98 | 0.9949 | 0.95 |
| 9 | 86.26 | 0.9917 | 1.32 |
| 10 | 85.94 | 0.9953 | 0.99 |
| 11 | 86.52 | 0.9887 | 1.49 |
| 12 | 86.35 | 0.9906 | 1.72 |
| 13 | 86.84 | 0.9851 | 1.90 |
| 14 | 87.05 | 0.9827 | 1.94 |
| 15 | 88.02 | 0.9719 | 2.40 |
| 16 | 87.98 | 0.9723 | 2.68 |
| **Averaged value** | 86.33 | 0.9909 | 1.20 |

**Table 3.** Cross-section under flexure. Global, small, random displacement (GSRD). Performance metrics.

| Subpopulations | Mean Cost | Accuracy | Robustness |
|---|---|---|---|
| 1 | 85.81 | 0.9969 | 0.51 |
| 2 | 86.08 | 0.9937 | 0.66 |
| 3 | 85.93 | 0.9955 | 0.62 |
| 4 | 85.73 | 0.9978 | 0.43 |
| 5 | 85.74 | 0.9976 | 0.45 |
| 6 | 85.74 | 0.9976 | 0.46 |
| 7 | 85.90 | 0.9958 | 1.04 |
| 8 | 85.76 | 0.9974 | 0.47 |
| 9 | 86.18 | 0.9926 | 1.96 |
| 10 | 85.79 | 0.9971 | 0.75 |
| 11 | 86.04 | 0.9941 | 1.08 |
| 12 | 85.80 | 0.9969 | 0.51 |
| 13 | 86.22 | 0.9921 | 1.22 |
| 14 | 85.89 | 0.9959 | 0.82 |
| 15 | 86.66 | 0.9871 | 2.22 |
| 16 | 86.33 | 0.9908 | 1.34 |
| **Averaged value** | 85.98 | 0.9949 | 0.91 |

**Table 4.** Cross-section under flexure. LSRD and GSRD. Performance metrics.

| Subpopulations | Mean Cost | Accuracy | Robustness |
|---|---|---|---|
| 1 | 85.81 | 0.9969 | 0.51 |
| 2 | 85.73 | 0.9977 | 0.43 |
| 3 | 85.71 | 0.9980 | 0.39 |
| 4 | 85.73 | 0.9978 | 0.43 |
| 5 | 85.73 | 0.9978 | 0.41 |
| 6 | 85.65 | 0.9988 | 0.27 |
| 7 | 85.71 | 0.9980 | 0.37 |
| 8 | 85.81 | 0.9968 | 0.51 |
| 9 | 85.99 | 0.9947 | 1.05 |
| 10 | 86.24 | 0.9918 | 1.43 |
| 11 | 85.85 | 0.9964 | 0.62 |
| 12 | 85.96 | 0.9951 | 0.99 |
| 13 | 86.06 | 0.9939 | 0.94 |
| 14 | 86.20 | 0.9923 | 1.20 |
| 15 | 86.32 | 0.9910 | 1.51 |
| 16 | 86.29 | 0.9914 | 1.38 |
| **Averaged value** | 85.93 | 0.9955 | 0.78 |

The method is able to obtain designs close to the global optimum as shown in Tables 2–4. When the method considers both LSRD and GSRD (Table 4), the minimum average value of the mean costs, the maximum average value of the accuracies, and the minimum average value of the standard deviations are obtained.

The performance metrics are also shown in Figures 5–7. It can be observed that the metrics tend to worsen in all cases when the number of subpopulations increases. This tendency is greater when LSRD is used alone and above eight subpopulations. The other cases have a lower tendency. The sensitivity of the metrics with respect to the number of subpopulations is reduced when LSRD and GSRD are taken into account at the same time.

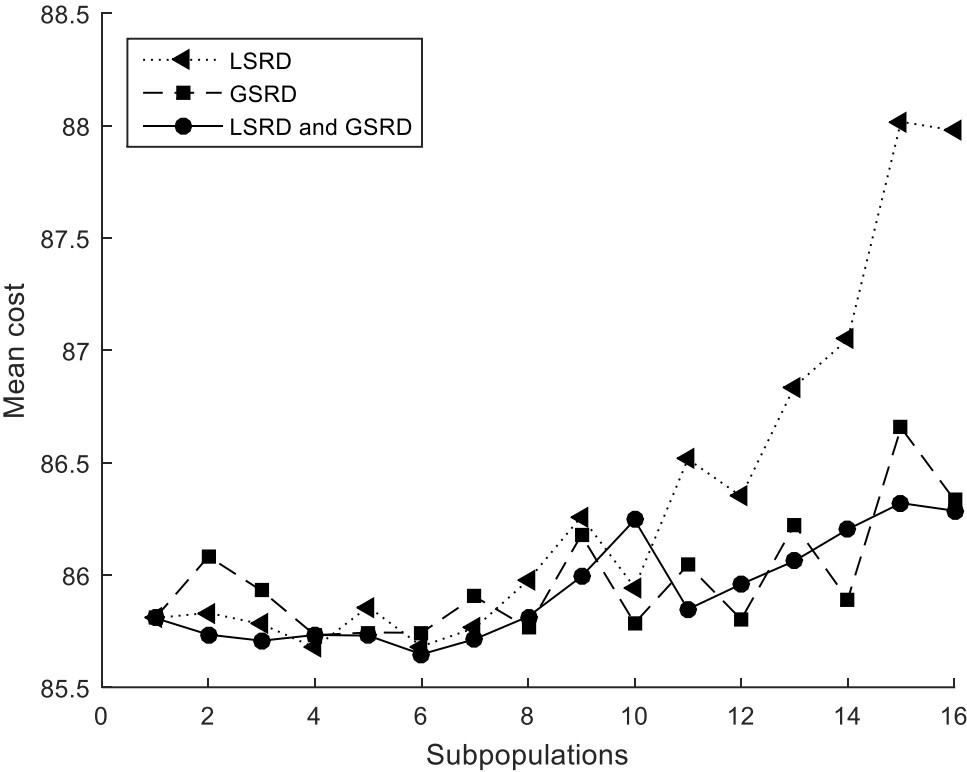

**Figure 5.** Cross-section under flexure. Mean cost.

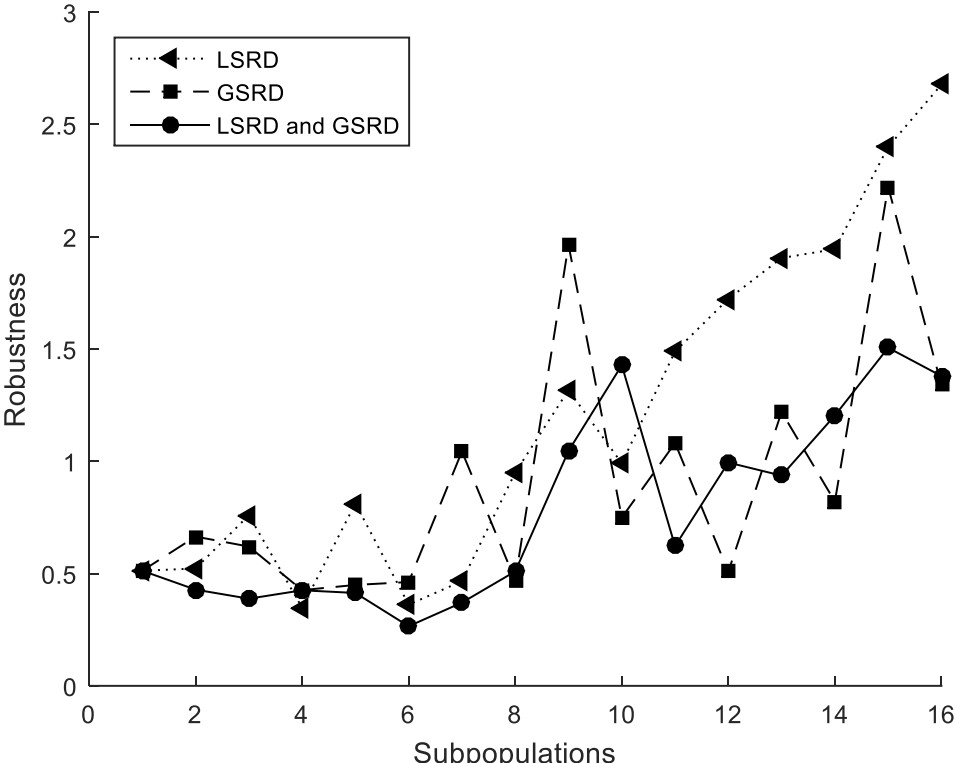

**Figure 6.** Cross-section under flexure. Robustness.

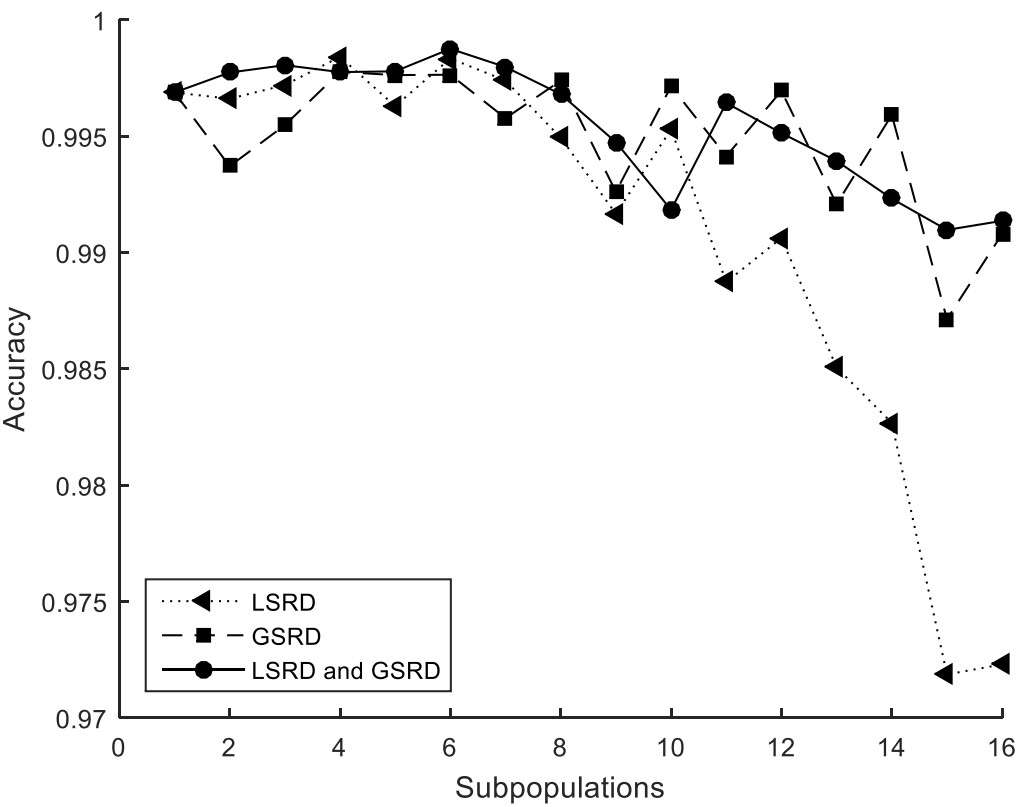

**Figure 7.** Cross-section under flexure. Accuracy.

### 3.2. Cross-Section under Biaxial Bending

A rectangular cross-section under biaxial bending [35] using MPFA is studied. The design loadings considered are $P_u$ = 200 kN, $M_{x,u}$ = 300 kNm and $M_{y,u}$ = 250 kNm. The strength of materials are $f_c'$ = 25.5 MPa for concrete and $f_y$ = 500 MPa for steel. The clear cover of reinforcement is 40 mm and the stirrup diameter is 10 mm. The cost is calculated using a price of $P_c$ = 100 €/m³ for concrete, of $P_s$ = 1.2 €/kg for steel and of $P_f$ = 15 €/m² for formwork. The length of form is $2b + 2h$. The optimization variable $\xi$ is constrained in the interval [−0.7, 0.7]; the angle of the neutral fiber $\theta$ in [0, $\pi/2$] rad; the width of the cross-section $b$ in [0.30, 0.50] m; the height $h$ in [0.30, 0.90] m; the reinforcement bar diameters $\varphi'_y$, $\varphi_y$, $\varphi'_x$, $\varphi_x$ in [10, 32] mm; the number of reinforcement bars $n'_y$, $n_y$ in [0, 10] and the number of reinforcement bars $n'_x$, $n_x$ in [2, 10]. The constraints (8) to (10), (12) to (15) and (16) are considered in this example. The parameters of Equation (16) are $C_1$ = 2 and $C_2$ = 5. The MPFA parameters are number of fireflies $NF$ = 300; maximum number of iterations $k_{max}$ = 1000; attractiveness at $r$ = 0 $\beta_0$ = 1; randomization parameter $\alpha_0$ = 1; absorption coefficient $\gamma_0$ = 10; $m$ = 2.0 and $t_c$ = 0.1.

The best solution obtained from 50 runs using MPFA with eight subpopulations is considered to be the optimal design solution. The optimal design solutions obtained considering ACI318 and EC2 standards are shown in Tables 5–8.

**Table 5.** Optimal design solutions with invariant section. EC2.

| | Case 1 | | Case 2 | Case 3 |
|---|---|---|---|---|
| | $\varphi'_y = \varphi_y = \varphi'_x = \varphi_x$ $n'_y = n_y$ $n'_x = n_x$ | | $\varphi'_y = \varphi_y$ $\varphi'_x = \varphi_x$ $n'_y = n_y$ $n'_x = n_x$ | |
| | Gil-Martín et al. [35] | MPFA | MPFA | MPFA |
| $z$ (mm) | - | 251.8 | 216.3 | 263.2 |
| $\theta$ (rad) | - | 1.0819 | 1.2200 | 1.2041 |
| $b$ (mm) | 400 | 400 | 400 | 400 |
| $h$ (mm) | 700 | 700 | 700 | 700 |
| $\varphi'_y$ (mm) | 14.4 | 20 | 10 | - |
| $n'_y$ | 6 | 5 | 5 | - |
| $\varphi_y$ (mm) | 14.4 | 20 | 10 | 14 |
| $n_y$ | 6 | 5 | 5 | 10 |
| $\varphi'_x$ (mm) | 14.4 | 20 | 32 | 10 |
| $n'_x$ | 8 | 2 | 2 | 2 |
| $\varphi_x$ (mm) | 14.4 | 20 | 32 | 14 |
| $n_x$ | 8 | 2 | 2 | 8 |
| $A_{st}$ (mm$^2$) | 4560.1 | 4398.2 | 4002.4 | 2928.0 |
| Relative $A_{st}$ (%) | 103.7 | 100.0 | 91.0 | 66.6 |
| Cost (€/m) | 103.5 | 102.0 | 98.3 | 88.3 |
| Relative cost (%) | 101.5 | 100.0 | 96.4 | 86.6 |

**Table 6.** Optimal design solutions with variable section. EC2.

| | Case 1 | | Case 2 | Case 3 |
|---|---|---|---|---|
| | $\varphi'_y = \varphi_y = \varphi'_x = \varphi_x$ $n'_y = n_y$ $n'_x = n_x$ | | $\varphi'_y = \varphi_y$ $\varphi'_x = \varphi_x$ $n'_y = n_y$ $n'_x = n_x$ | |
| | Gil-Martín et al. [35] | MPFA | MPFA | MPFA |
| $z$(mm) | - | 269.9 | 250.4 | 280.3 |
| $\theta$(rad) | - | 0.8660 | 1.0035 | 1.1029 |
| $b$(mm) | - | 495 | 460 | 465 |
| $h$(mm) | - | 575 | 620 | 615 |
| $\varphi'_y$(mm) | - | 12 | 12 | - |
| $n'_y$ | - | 8 | 2 | - |
| $\varphi_y$(mm) | - | 12 | 12 | 10 |
| $n_y$ | - | 8 | 2 | 10 |
| $\varphi'_x$(mm) | - | 12 | 32 | 10 |
| $n'_x$ | - | 10 | 2 | 2 |
| $\varphi_x$(mm) | - | 12 | 32 | 16 |
| $n_x$ | - | 10 | 2 | 9 |
| $A_{st}$(mm$^2$) | - | 4070.2 | 3667.0 | 2752.0 |
| Relative $A_{st}$ (%) | - | 100.0 | 90.1 | 67.6 |
| Cost (€/m) | - | 98.5 | 95.1 | 86.6 |
| Relative cost (%) | - | 100.0 | 96.5 | 87.9 |

**Table 7.** Optimal design solutions with invariant section. ACI318.

| | **Case 1** | | **Case 2** | **Case 3** |
|---|---|---|---|---|
| | $\varphi'_y = \varphi_y = \varphi'_x = \varphi_x$ <br> $n'_y = n_y$ <br> $n'_x = n_x$ | | $\varphi'_y = \varphi_y$ <br> $\varphi'_x = \varphi_x$ <br> $n'_y = n_y$ <br> $n'_x = n_x$ | |
| | Gil-Martín et al. [35] | MPFA | MPFA | MPFA |
| $z$(mm) | - | 222.7 | 203.2 | 242.1 |
| $\theta$(rad) | - | 1.1585 | 1.2390 | 1.1597 |
| $b$(mm) | - | 400 | 400 | 400 |
| $h$(mm) | - | 700 | 700 | 700 |
| $\varphi'_y$(mm) | - | 16 | 12 | 10 |
| $n'_y$ | - | 9 | 9 | 4 |
| $\varphi_y$(mm) | - | 16 | 12 | 16 |
| $n_y$ | - | 9 | 9 | 10 |
| $\varphi'_x$(mm) | - | 16 | 25 | 12 |
| $n'_x$ | - | 4 | 3 | 3 |
| $\varphi_x$(mm) | - | 16 | 25 | 16 |
| $n_x$ | - | 4 | 3 | 5 |
| $A_{st}$(mm$^2$) | - | 5217.6 | 4978.7 | 3669.4 |
| Relative $A_{st}$ (%) | - | 100.0 | 95.4 | 70.3 |
| Cost (€/m) | - | 109.6 | 107.4 | 95.2 |
| Relative cost (%) | - | 100.0 | 98.0 | 86.9 |

**Table 8.** Optimal design solutions with variable section. ACI318.

| | **Case 1** | | **Case 2** | **Case 3** |
|---|---|---|---|---|
| | $\varphi'_y = \varphi_y = \varphi'_x = \varphi_x$ <br> $n'_y = n_y$ <br> $n'_x = n_x$ | | $\varphi'_y = \varphi_y$ <br> $\varphi'_x = \varphi_x$ <br> $n'_y = n_y$ <br> $n'_x = n_x$ | |
| | Gil-Martín et al. [35] | MPFA | MPFA | MPFA |
| $z$(mm) | - | 251.8 | 243.7 | 264.6 |
| $\theta$(rad) | - | 0.8095 | 0.9426 | 0.8786 |
| $b$(mm) | - | 500 | 485 | 485 |
| $h$(mm) | - | 565 | 605 | 615 |
| $\varphi'_y$(mm) | - | 14 | 12 | 10 |
| $n'_y$ | - | 7 | 6 | 4 |
| $\varphi_y$(mm) | - | 14 | 12 | 14 |
| $n_y$ | - | 7 | 6 | 9 |
| $\varphi'_x$(mm) | - | 14 | 25 | 10 |
| $n'_x$ | - | 8 | 3 | 3 |
| $\varphi_x$(mm) | - | 14 | 25 | 14 |
| $n_x$ | - | 8 | 3 | 7 |
| $A_{st}$(mm$^2$) | - | 4618.1 | 4300.6 | 3012.8 |
| Relative $A_{st}$ (%) | - | 100.0 | 93.1 | 65.2 |
| Cost (€/m) | - | 103.2 | 102.1 | 90.9 |
| Relative cost (%) | - | 100.0 | 98.9 | 88.1 |

The average time of 50 runs, considering ACI318 standard and using MPFA with different numbers of subpopulations, was obtained. The speed-up and efficiency are shown in Figures 8 and 9.

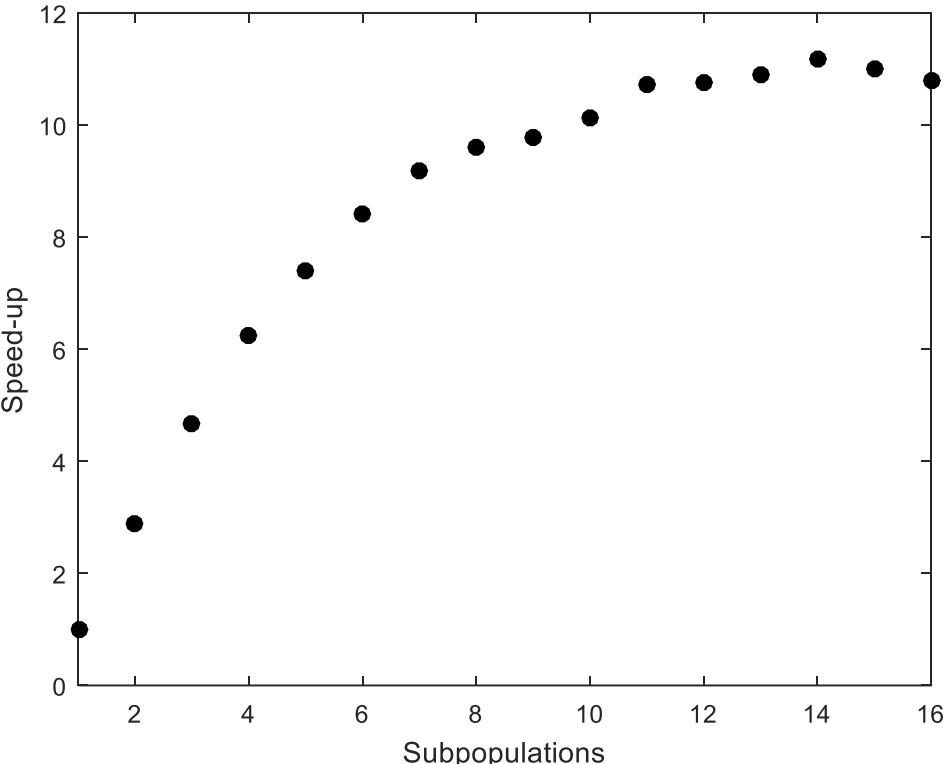

**Figure 8.** Cross-section under biaxial bending. Speed-up.

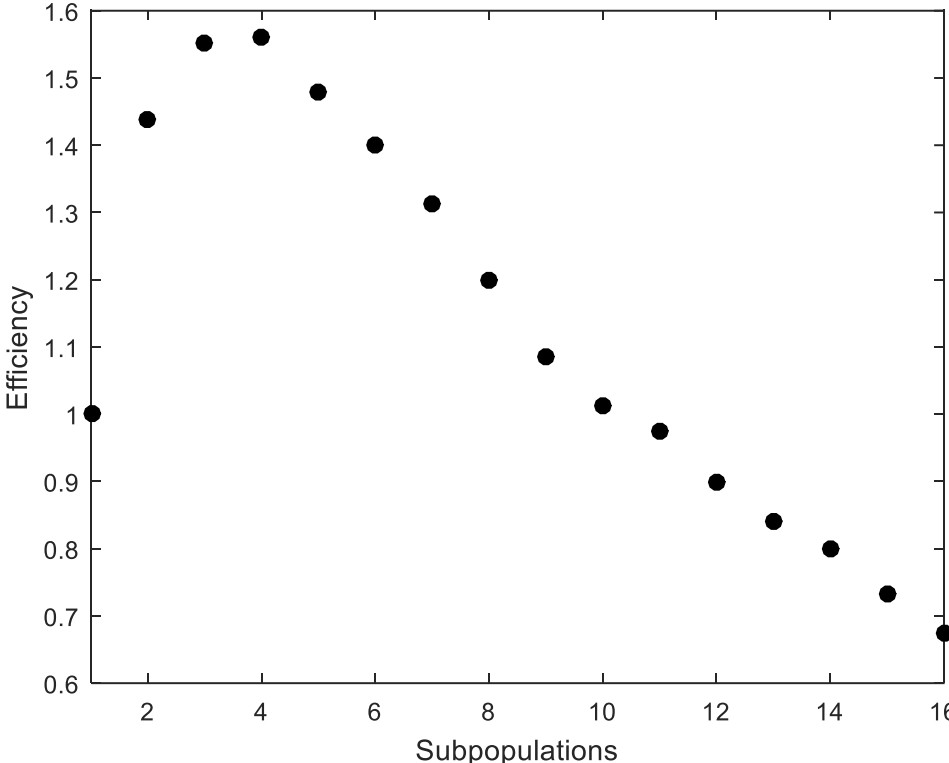

**Figure 9.** Cross-section under biaxial bending. Efficiency.

As in the previous example, the parallelization produces satisfactory results (Figure 8), although the efficiency quickly drops with more than four processors (Figure 9).

The mean value, accuracy (reference value of Cost = 90.9, from Table 8), and robustness (standard deviation) from 50 runs considering ACI318 standard and using MPFA with a

different number of subpopulations were obtained and shown in Tables 9–11, considering only LSRD, only GSRD, and both LSRD and GSRD, respectively. For comparison purposes, the average values of the performance metrics were also obtained, and they are shown in the last row of Tables 9–11.

**Table 9.** Cross-section under biaxial bending. LSRD. Performance metrics.

| Subpopulations | Mean Cost | Accuracy | Robustness |
|:---:|:---:|:---:|:---:|
| 1 | 91.92 | 0.9889 | 0.71 |
| 2 | 92.02 | 0.9878 | 0.62 |
| 3 | 91.91 | 0.9890 | 0.58 |
| 4 | 91.75 | 0.9907 | 0.50 |
| 5 | 92.10 | 0.9869 | 0.91 |
| 6 | 92.24 | 0.9855 | 0.69 |
| 7 | 92.28 | 0.9850 | 0.77 |
| 8 | 92.39 | 0.9839 | 0.92 |
| 9 | 92.68 | 0.9808 | 1.06 |
| 10 | 92.64 | 0.9812 | 1.05 |
| 11 | 92.71 | 0.9805 | 1.29 |
| 12 | 92.69 | 0.9807 | 1.17 |
| 13 | 92.68 | 0.9808 | 0.99 |
| 14 | 92.92 | 0.9783 | 1.07 |
| 15 | 92.64 | 0.9812 | 0.93 |
| 16 | 93.05 | 0.9769 | 1.21 |
| **Averaged value** | 92.41 | 0.9836 | 0.91 |

**Table 10.** Cross-section under biaxial bending. GSRD. Performance metrics.

| Subpopulations | Mean Cost | Accuracy | Robustness |
|:---:|:---:|:---:|:---:|
| 1 | 91.92 | 0.9889 | 0.71 |
| 2 | 91.68 | 0.9915 | 0.55 |
| 3 | 91.87 | 0.9894 | 0.69 |
| 4 | 91.87 | 0.9895 | 0.60 |
| 5 | 91.96 | 0.9885 | 0.64 |
| 6 | 91.97 | 0.9884 | 0.61 |
| 7 | 91.98 | 0.9882 | 0.85 |
| 8 | 92.26 | 0.9853 | 0.76 |
| 9 | 92.17 | 0.9862 | 0.74 |
| 10 | 92.39 | 0.9839 | 0.95 |
| 11 | 92.11 | 0.9869 | 0.73 |
| 12 | 92.45 | 0.9832 | 0.93 |
| 13 | 92.25 | 0.9854 | 0.88 |
| 14 | 92.51 | 0.9826 | 0.76 |
| 15 | 92.45 | 0.9832 | 0.81 |
| 16 | 92.59 | 0.9817 | 0.89 |
| **Averaged value** | 92.15 | 0.9864 | 0.76 |

**Table 11.** Cross-section under biaxial bending. LSRD and GSRD. Performance metrics.

| Subpopulations | Mean Cost | Accuracy | Robustness |
|---|---|---|---|
| 1 | 91.92 | 0.9889 | 0.71 |
| 2 | 91.87 | 0.9895 | 0.60 |
| 3 | 91.87 | 0.9895 | 0.62 |
| 4 | 91.97 | 0.9884 | 0.71 |
| 5 | 92.09 | 0.9871 | 0.77 |
| 6 | 92.02 | 0.9878 | 0.77 |
| 7 | 92.20 | 0.9859 | 0.80 |
| 8 | 92.25 | 0.9854 | 0.82 |
| 9 | 92.17 | 0.9862 | 0.79 |
| 10 | 92.20 | 0.9859 | 0.83 |
| 11 | 92.27 | 0.9852 | 0.80 |
| 12 | 92.25 | 0.9853 | 0.79 |
| 13 | 92.19 | 0.9860 | 0.82 |
| 14 | 92.47 | 0.9831 | 0.79 |
| 15 | 92.28 | 0.9851 | 0.89 |
| 16 | 92.49 | 0.9828 | 0.86 |
| **Averaged value** | 92.16 | 0.9864 | 0.77 |

The method is able to obtain designs close to the global optimum as shown in Tables 9–11. When the method considers only GSRD (Table 10), the minimum average value of the mean costs, the maximum average value of the accuracies, and the minimum average value of the standard deviations are obtained. Very similar values, but somewhat higher, are obtained when LSRD and GSRD (Table 11) are taken into account together.

The performance metrics are also shown in Figures 10–12. It can be observed that the metrics tend to worsen in all cases when the number of subpopulations increases. This tendency is greater when LSRD is used alone and above six subpopulations. The other cases have a lower tendency. The sensitivity of the metrics with respect to the number of subpopulations is reduced when LSRD and GSRD are taken into account at the same time.

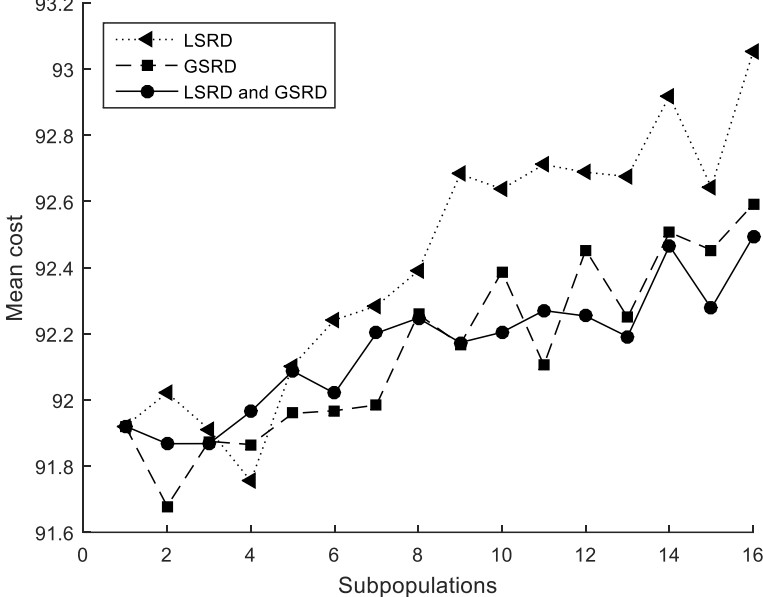

**Figure 10.** Cross-section under biaxial bending. Mean cost.

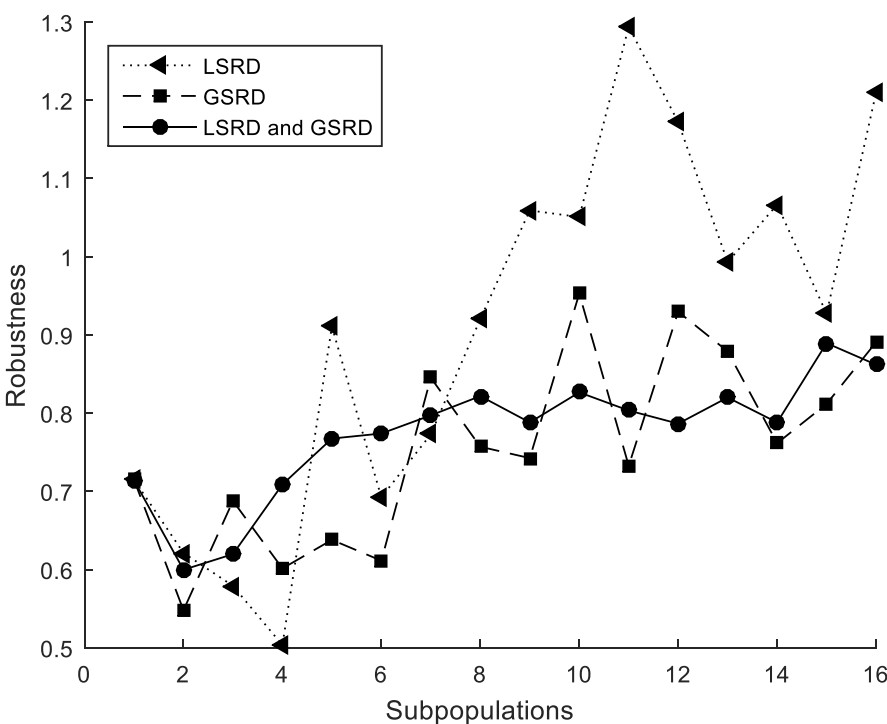

**Figure 11.** Cross-section under biaxial bending. Robustness.

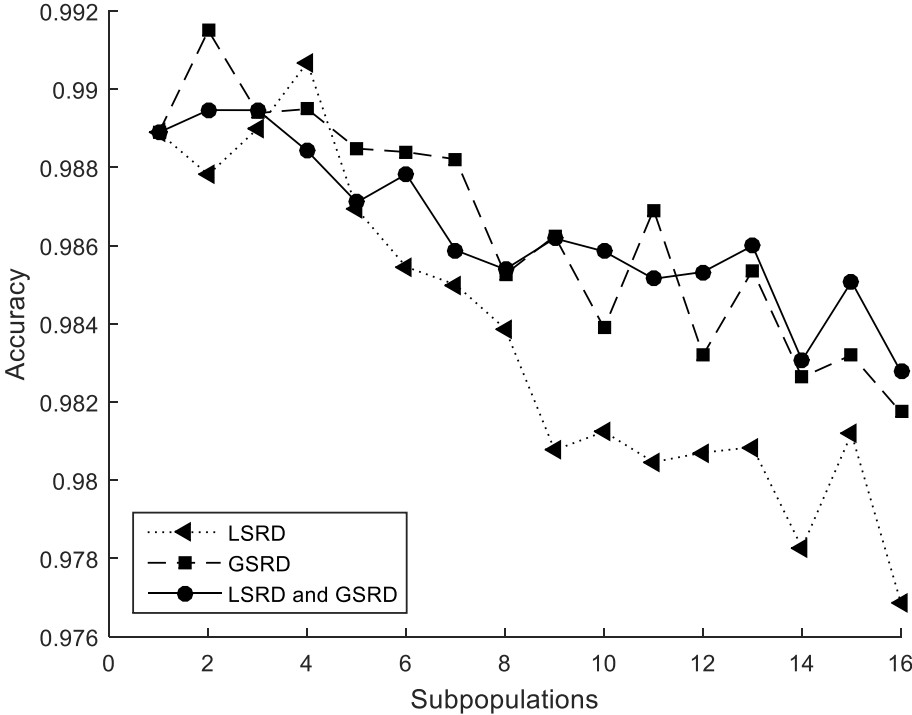

**Figure 12.** Cross-section under biaxial bending. Accuracy.

## 4. Conclusions and Final Remarks

A simple formulation for the optimal design of reinforced concrete sections under compression and biaxial bending was established in a previous work by the authors. In that work, to verify that the formulation could work well in different types of metaheuristic algorithms, a genetic algorithm and two firefly algorithms were used to obtain the solutions to two problems under the conditions of two different design codes (EC2 and ACI318).

As meta-heuristic algorithms have the common disadvantage of needing to evaluate the objective function many times during the exploration phase, the time required to obtain the solution was extensive.

For this reason, the authors studied and implemented a simple parallel computing strategy and the corresponding modifications and adjustments in the FA, which is explained in detail in this work, with the aim of: (i) reducing the necessary calculation time and (ii) avoiding any lack of precision regardless of the number of parallel processes adopted.

In accordance with these two objectives, the two examples show that:

(i)    The speed-up increases as more parallel processes are considered. The trend is almost linear up to six parallel processes. From six onwards, there is no significant increase. It can also be seen that the efficiency quickly drops with more than four parallel processes. These results depend on the computer used to run the calculations but not on the proposed design method. To achieve better results, using another computer whose architecture allows more processes in parallel would be sufficient.

(ii)   The method achieves designs close to the global optimum despite the number of parallel processes considered. Small random displacements (LSRD and GSRD) have proven to be essential to avoiding the bias produced by the migration between subpopulations. LSRD shows its effect when there are few subpopulations of many individuals. In contrast, GSRD shows its effect when there are many subpopulations with few individuals. The combined use of LSRD and GSRD slightly improves the results and reduces their sensitivity in relation to the number of parallel processes considered. It should be noted that the proposed method was tested considering only up to 16 parallel processes (or 16 subpopulations). More research should be done considering more parallel processes. To facilitate this task for other potential researchers or for other uses, the authors included the MATLAB®code for the complete method in the Appendix A.

This cross-sectional design method can be integrated into a more general design scheme for three-dimensional reinforced concrete structures by making the necessary changes and adjustments. To facilitate this integration, it was previously designed to be able to solve different load situations, as shown in the two examples included in this work.

**Supplementary Materials:** The following are available online at https://www.mdpi.com/2076-3417/11/5/2076/s1.

**Author Contributions:** Conceptualization, A.T. and G.S.-O.; methodology and algorithm implementation, G.S.-O.; checking the results, A.T.; writing-original draft preparation, G.S.-O.; writing-review and editing, A.T. All authors have read and agreed to the published version of the manuscript.

**Funding:** This research received no external funding.

**Institutional Review Board Statement:** Not applicable.

**Informed Consent Statement:** Not applicable.

**Data Availability Statement:** The data presented in this study are available on request from the corresponding author.

**Conflicts of Interest:** The authors declare that they have no known competing financial interests or personal relationships that could have appeared to influence the work reported in this paper.

**Appendix A**

To execute the following MATLAB® functions, it is necessary to select an appropriate number of workers in the "Parallel preferences" menu, "Parallel tool" option before execution. There, the user must set the "Preferred number of workers in a parallel survey" to a value greater than the number of subpopulations considered. This value is related to the number of processor cores.

Every function included here must be located in an independent .m file.

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
