# Peer review of "Optimization of Reinforced Concrete Sections under Compression and Biaxial Bending by Using a Parallel Firefly Algorithm"

_applsci, doi:10.3390/app11052076_

Round 1
Reviewer 1 Report
This paper presents a parallel firefly algorithm to optimise reinforced concrete sections under compression and biaxial bending. Generally, the authors achieve their goals with the present algorithm. This paper has sound scientific value and good organisation. After a minor revision, this paper deserves to be published with a good contribution to the field.
1) Please slightly modify the abstract to focus on the scientific contribution of this work, instead of talking too much about the authors’ previous work.
2) The equations are not in the same size with other texts. Please consider modifying.
3) It is recommended to include a brief introduction on the analysis object in the introduction, like ‘the reinforced concrete structures have been widely used in the building [*1], road [*2], rail [*3] and marine [*4] engineering…’, which is able to indicate the necessity of this work. The number of references is not quite enough for a numerical simulation paper.
[*1] Kunnath, S. K., A. M. Reinhorn, and J. F. Abel. "A computational tool for evaluation of seismic performance of reinforced concrete buildings." Computers & structures 41.1 (1991): 157-173.
[*2] Carbonell, Alfonso, Fernando González-Vidosa, and Víctor Yepes. "Design of reinforced concrete road vaults by heuristic optimization." Advances in Engineering Software 42.4 (2011): 151-159.
[*3] Song Y, Wang Z, Liu Z, Wang R. A spatial coupling model to study dynamic performance of pantograph-catenary with vehicle-track excitation. Mech Syst Signal Process 2021;151:107336.
[*4] Kondratova, I. L., P. Montes, and T. W. Bremner. "Natural marine exposure results for reinforced concrete slabs with corrosion inhibitors." Cement and Concrete Composites 25.4-5 (2003): 483-490.
Reviewer 2 Report
The present papers deals with an optimization analysis of reinforced concrete sections under compression through a firefly algorithm. The topic is scientifically interesting and the section are well organized. The reviewers wish to address few comments: the theoretical background should be enriched with other optimization works on structural problems, e.g. a novel PDSO algorithm has been developed in the following work
(2019), ”Design of a noise reduction passive control system based on viscoelastic multilayered plate using PDSO”, Mechanical Systems and Signal Processing 123: 153-173 DOI: http://dx.doi.org/10.1016/j.ymssp.2019.01.011
Moreover the reviewer suggests to rewrite the Conclusions section listing the main findings of the work, in order to improve the readability of the section.
Except for these minor comments, the manuscript could be considered for publication.
Reviewer 3 Report
The paper discusses application of firefly algorithm to the problem of optimization of the size and reinforcement of concrete blocks. The algorithm was applied in previous paper by the authors, and its parallel version is developed and investigated concerning its efficiency in the present work. In spite of rather simplified calculation of strength of the structure in the paper, the computer optimization of complex engineering structures is relevant. The paper is clear written and can be recommended for publication after addressing the following issues. A brief explanation of the optimization problem including constrains and objective in several sentences should be included at the end of Introduction or in Section 2.1. A brief qualitative explanation of the principal of the firefly algorithm is required in the same place. Also check the numeration of sections.
